# Enhancing Adversarial Transferability Through Exploiting Multiple Randomized Trajectories for Better Global Guidance

## Abstract

Deep neural networks are well-known for their vulnerability to adversarial examples, particularly demonstrating poor performance in white-box attack settings. However, most white-box attack methods heavily depend on the target model, and the adversarial samples often get trapped in local optima, leading to limited adversarial transferability. Although techniques such as momentum, variance reduction, and gradient penalty mitigate overfitting by combining historical information with information from local regions around adversarial examples, still, much of the global loss landscape remains unexplored, hindering further performance improvements.

In this work, we find that random initialization influences the optimization of adversarial examples, making them converge at multiple local optima, leaving the rest of the loss landscape unexplored. Based on this insight, we propose two strategies: randomized global initialization and dual examples. These strategies utilize multiple optimization trajectories to capture global optimization directions, enhancing adversarial transferability. Our approach integrates seamlessly with existing adversarial attack methods and significantly improves transferability, as demonstrated by empirical evaluations on the standard ImageNet dataset.

## 1 Introduction

Adversarial examples, which involve subtle perturbation to benign samples that can mislead deep neural networks (DNNs), have garnered considerable attention in recent years (Szegedy et al., 2013; Goodfellow et al., 2015; Wang et al., 2019). These examples underscore the susceptibility of DNNs and raise significant security issues across various fields, including autonomous driving (Cao et al., 2019; Nesti et al., 2022; Girdhar et al., 2023), facial authentication (Chen et al., 2017; 2019; Joos et al., 2022), and object detection (Li et al., 2021; Nezami et al., 2021; Zhang and Wang, 2019), among others. The investigation into adversarial examples has led to extensive research focused on enhancing the robustness (Madry et al., 2018; Shafahi et al., 2019; Zheng et al., 2020; Jia et al., 2022) and comprehension (Shumailov et al., 2019) of DNNs. In summary, adversarial examples have become essential for identifying vulnerabilities and improving the robustness of DNNs.

Without any knowledge about the architecture, parameters, or logits of remote victim models used in real-world applications, attackers often use local surrogate models to generate adversarial examples that can deceive these victim models, a method known as transfer adversarial attacks. Various methods have been developed to improve adversarial transferability. These methods include gradient-based attacks (Dong et al., 2018; Wang and He, 2021; Wang et al., 2021b; Ge et al., 2023), input transformation-based attacks (Xie et al., 2019; Dong et al., 2019; Wang et al., 2021a; 2024; 2023b; Wang and Yin, 2023), and model-related attacks (Liu et al., 2017; Xiong et al., 2022; Gubri et al., 2022; Wang et al., 2023a;c).

Gradient-based methods form the foundation of various attack techniques, including input transformation and model-related approaches. Goodfellow et al. (2015) introduced FGSM, using gradient ascent for adversarial transferability, while Kurakin et al. (2018) enhanced this

with iterative steps. However, adversarial optimization often stagnates in local maxima when relying solely on gradients. Techniques like momentum (Dong et al., 2018), Nesterov (Lin et al., 2020), variance reduction (Wang et al., 2021b; Wang and He, 2021), and gradient norm penalization (Ge et al., 2023) have improved transferability. Input transformation-based methods, by incorporating input diversity at each step, further enhance generalization and attack performance. These methods underscore the importance of exploring loss landscapes for better global guidance. However, as input transformations often require predefined transformations and involve higher memory and computation costs, a natural question arises: Can broader loss landscape exploration be integrated into the iterative attack process of gradient-based methods?

Unlike previous methods that focus on exploring regions around adversarial examples, our approach broadens the exploration by navigating around benign samples. Specifically, we investigate the often-overlooked role of initialization in adversarial attacks. While initialization may not significantly impact performance, it can lead adversarial optimization to multiple local optima. Based on this finding, we propose two simple yet effective strategies—randomized global initialization and dual examples—to leverage the entire loss landscape around benign samples, thereby enhancing global guidance and improving adversarial transferability.

Our contributions are summarized as follows,

- Using t-SNE to project the optimization trajectory of adversarial examples into a visualizable latent space, we empirically validate that random initialization can guide adversarial optimization to multiple local optima without compromising adversarial transferability.

- We propose two simple yet effective strategies—randomized global initialization and dual examples—to enhance adversarial transferability by usling multiple trajectories to explore broader loss landscapes, utilizing multiple continuous optimization trajectories to capture global information.

- Extensive experiments on the ImageNet-1K dataset demonstrate the effectiveness of our approach, achieving state-of-the-art performance in gradient-based transferable attack settings.

## 2 RELATED WORK

### 2.1 ADVERSARIAL ATTACK AND ADVERSARIAL TRANSFERABILITY

Since Szegedy et al. (2013) uncovered the vulnerability of DNNs to adversarial examples, numerous adversarial attacks have been proposed, including 1) *white-box attacks*: the attacker has the full knowledge of the victim model (Goodfellow et al., 2015; Moosavi-Dezfooli et al., 2016; Carlini and Wagner, 2017), *e.g.*, architecture, logits. 2) *black-box attacks*: the attacker has no prior information of the victim model. It is often impossible to access information about the target victim model in real-world scenarios, necessitating black-box attack techniques. Existing black-box attacks can be grouped into three classes: score-based (Andriushchenko et al., 2020; Yatsura et al., 2021), decision-based (Li et al., 2022; Chen et al., 2020; Wang et al., 2022b), and transfer-based (Dong et al., 2018; Lin et al., 2020; Wang et al., 2021a) attacks. Score-based and decision-based attacks typically require a significant number of queries on the victim model, while transfer-based attacks adopt the adversarial examples generated on surrogate models to fool different victim models. This makes transfer-based attacks more computationally efficient and better suited for real-world applications. Hence, we focus on transfer-based attacks. Numerous researchers have devised strategies to enhance adversarial transferability, concentrating mainly on three approaches: iterative gradient-based optimization, input transformation-based methods, and model-related techniques.

**Gradient-based optimization methods**. I-FGSM (Kurakin et al., 2018) extends FGSM (Goodfellow et al., 2015) into an iterative version to substantially enhance the attack effectiveness under the white-box setting but exhibits poor transferability. MI-FGSM (Dong et al., 2018) incorporates momentum to improve adversarial transferability, while NI-FGSM (Lin et al., 2020) applies Nesterov momentum for optimization acceleration.

PI-FGSM (Gao et al., 2020) recycles the clipped adversarial perturbation to the neighbor pixels to enhance the transferability. VMI-FGSM (Wang and He, 2021) adjusts the gradient based on the gradient variance of the previous iteration to stabilize the update direction. EMI-FGSM (Wang et al., 2021b) enhances the momentum by averaging the gradient of data points sampled from the optimization direction. GIMI-FGSM (Wang et al., 2022a) initializes the momentum by running the attacks in several iterations for gradient pre-convergence.

**Input transformation methods**. Input transformation-based attacks have shown great effectiveness in improving transferability. For instance, diverse input method (DIM) (Xie et al., 2019) resizes the input image to a random size, which is then padded to a fixed size for gradient calculation. TIM (Dong et al., 2019) adopts Gaussian smooth on the gradient to approximate the average gradient of a set of translated images to update the adversary. Scale-invariant method (SIM) (Lin et al., 2020) calculates the gradient on a collection of scaled images. *Admix* (Wang et al., 2021a) incorporates a fraction of images from other categories into the inputs to generate multiple images for gradient calculation. SSA (Long et al., 2022) randomly transforms the image in the frequency domain to craft more transferable adversarial examples.

**Model-related methods**. Liu *et al.* (Liu et al., 2017) initially discovered that an ensemble attack, which generates adversarial examples on multiple models, can result in better transferability. Li *et al.* (Li et al., 2020) simultaneously attack several ghost networks, which are generated by adding dropout layers to the surrogate model. Xiong *et al.* (Xiong et al., 2022) minimize the gradient variance across different models to enhance ensemble attacks. Gubri *et al.* (Gubri et al., 2022) train the model with a high learning rate to produce multiple models and attack them sequentially to improve existing attacks' transferability.

## 2.2 ADVERSARIAL DEFENSE

To mitigate the threat of adversarial attacks, a variety of defense methods have been proposed, including adversarial training Goodfellow et al. (2015); Zhang et al. (2019); Wang et al. (2020), input pre-processing Guo et al. (2018), certified defense Cohen et al. (2019), *etc.* For example, Liao *et al.* Liao et al. (2018) proposes a high-level representation guided denoiser (HGD) to purify the adversarial examples. Madry *et al.* Madry et al. (2018) introduces an adversarial training method (AT) that utilizes PGD adversarial examples to train models, aiming to enhance their adversarial robustness. Wong *et al.* Wong et al. (2020) employ random initialization in FGSM adversarial training, leading to Fast Adversarial Training (FAT), which achieves accelerated training and improved adversarial robustness comparable to PGD training. Cohen *et al.* Cohen et al. (2019) propose a random smoothing technique (RS) to provide the model with certified robustness against the adversarial examples. Naseer *et al.* Naseer et al. (2020) design a neural representation purifier (NRP) to remove harmful perturbations of images.

## 3 METHODOLOGY

### 3.1 TRACING THE OPTIMIZATION TRAJECTORY IN RANDOMIZED ADVERSARIAL ATTACKS

Initialization techniques (e.g., random start, Xavier (Glorot and Bengio, 2010), Kaiming (He et al., 2015)) are widely recognized for expediting convergence in optimization problems. While prior studies (Lin et al., 2020; Wang et al., 2021a; 2023b) have drawn empirical connections between neural network training and adversarial example generation in terms of generalization, the role of initialization in adversarial contexts remains underexplored. The first work addressing this is GIMI-FGSM (Wang et al., 2022a), which initializes momentum with a pre-computed value. In this work, we conduct a more detailed investigation into the impact of initialization on adversarial example generation.

In particular, we explore the initialization strategy of randomly initializing the adversarial perturbation. We evaluate three attack methods: I-FGSM (Kurakin et al., 2018), VMI-FGSM (Wang et al., 2021b), and GIMI-FGSM (Wang et al., 2022a). To test this, we generate 1,000 adversarial examples targeting the ResNet-18 (He et al., 2016) surrogate model and assess their transferability across six models: ResNet-101 (He et al., 2016),

ResNeXt-50 (Xie et al., 2017), DenseNet-121 (Huang et al., 2017), MobileNet (Howard et al., 2017), ViT (Dosovitskiy et al., 2020), and Swin (Liu et al., 2021). We present the results of different random start experiments in fig. 1. Our results demonstrate that attacks initialized with different random perturbations perform comparably to each other, where the maximum difference between attack success rates is only 1.6%, which is indicated by the surrounded shadow area of each line. However, the question remains: what does change?

To gain deeper insights into how perturbation initialization influences the dynamics of adversarial attacks, we propose using $t$-distributed Stochastic Neighbor Embedding (t-SNE) to project the optimization trajectory of adversarial examples into a latent space for visualization. Specifically, for a benign sample $x$, we generate a series of adversarial examples $x_1, x_2, \ldots, x_{20}$ using different attack methods with increasing numbers of steps $t = 1, 2, \ldots, 20$ with fixed step size. To obtain the projections $z_1, z_2, \ldots, z_{20}$ in the latent space, we optimize the following loss function:

$$\mathcal{L} = \sum_{i \neq j} P_{ij} \log \left( \frac{P_{ij}}{Q_{ij}} \right), \qquad (1)$$

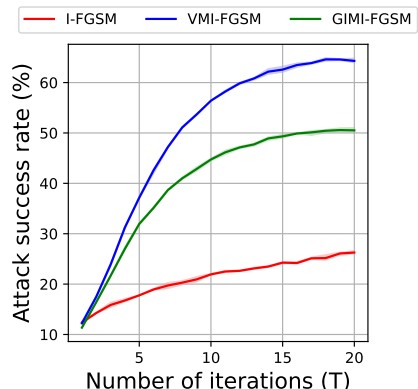

Figure 1: Results of the attack success rate (ASR) versus the epoch for I-FGSM, VMI-FGSM, and GIMI-FGSM with fixed step size.

where $P_{ij}$ is the similarity between points $x_i$ and $x_j$ in the high-dimensional space, modeled using a Gaussian kernel, and $Q_{ij}$ is the similarity between their projections $z_i$ and $z_j$ in the latent space, modeled using a Student's $t$-distribution. By minimizing $\mathcal{L}$ through gradient descent, we iteratively adjust $\{z_t\}_{t=1}^{20}$ to preserve the local structure of the data. This approach allows us to visualize the optimization trajectory of adversarial examples in the latent space, reflecting their relationships in the original high-dimensional space.

We present the results in fig. 2, where it becomes clear that for all three methods, different random initializations lead the optimization of the same adversarial example to converge to distinct local optima. Specifically, while VMI-FGSM employs variance reduction to stabilize the optimization trajectory compared to I-FGSM, it still fails to reach a consistent optimum across different random initializations. Even with global momentum pre-computed for momentum initialization, GIMI-FGSM does not achieve a unified global direction. Besides, by examining the trajectories of different attacks, we observe that even with the same step size and number of optimization steps, each attack pushes the adversarial example to different distances from the benign sample. Notably, I-FGSM converges the fastest, while VMI-FGSM drives the adversarial example the farthest from the benign sample.

### 3.2 LEVERAGING MULTIPLE TRAJECTORIES TO ENHANCE THE ADVERSARIAL TRANSFERABILITY

From the visualization results, we observe that significant portions of the loss landscape remain under-explored, causing the optimization of adversarial examples to become easily trapped in multiple local optima around benign samples. To enhance adversarial transferability, we propose two strategies: randomized global initialization and dual example generation. These strategies leverage multiple parallel trajectories to explore the loss landscape more comprehensively.

**Randomized global initialization**. Building on the design of GIMI-FGSM, which initializes momentum using pre-computed global guidance, we take a further step to address the challenge of accurately capturing true global momentum. Pre-computation is complicated by the presence of multiple local optima near the initial benign samples. For instance, as shown in Figure 2 , running GIMI-FGSM from different random starting points often causes adversarial examples to converge to distinct local optima, which can hinder adversarial transferability, especially with a large number of iterations.

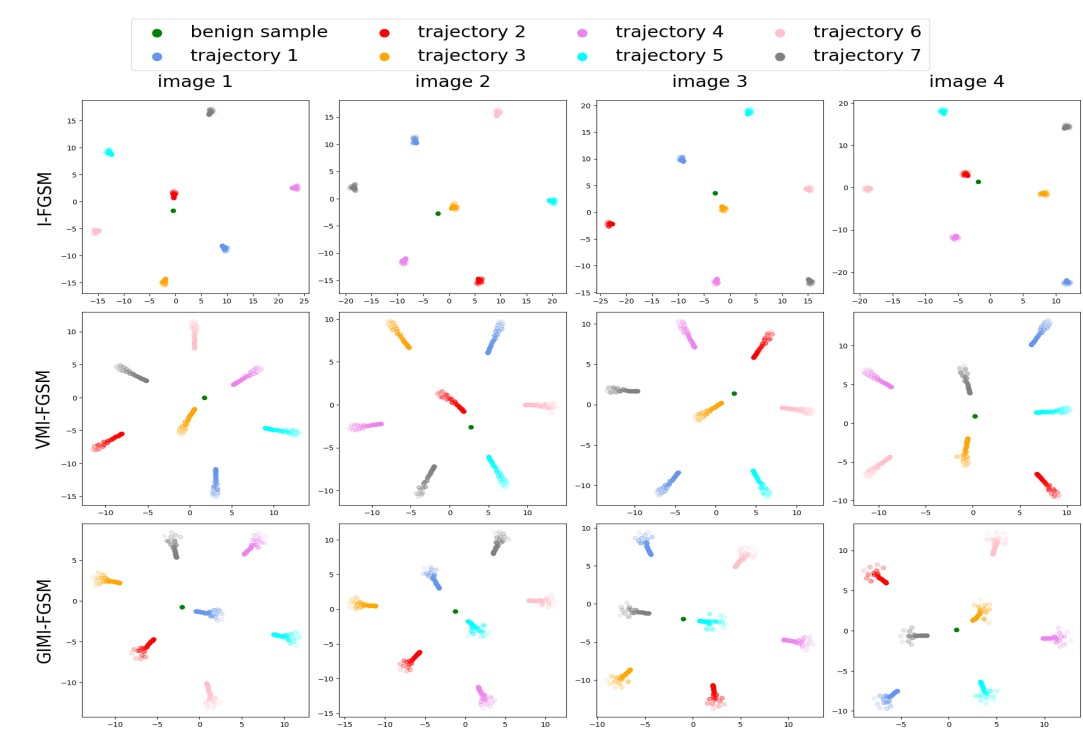

Figure 2: Visualization of I-FGSM, VMI-FGSM, and GIMI-FGSM with different random starts. The 20-step optimization trajectories are projected into the latent space, where the transparency indicates the step number: more transparent dots correspond to later steps.

---

**Algorithm 1** Boosting the adversarial transferability of MI-FGSM with RGI and DE.

---

**Input:** The neural network $f(\cdot)$, benign sample $x$ with the ground truth $y$, loss function $\mathcal{L}$, number of iterations $T$, number of dual examples $K$, momentum decay factor $\gamma$, number of samples used for computing the randomized global momentum $N$, increasing scheduled step size sequence $\{\alpha_t\}_{t=1}^T$.

**Output:** The adversarial perturbation.

1: Initialize $\{\delta_{k,0}^{dual}\}_{k=1}^K$ using the random initialization, and $\delta_0 = 0$
2: Initialize the momentum $m_0$ with 0
3: **for** $n = 1$ to $N$ **do**
4:     Initialize the momentum $m_{n,0} = 0$, and randomly initialize $\delta_{n,0}$
5:     **for** $t = 1$ to $T'$ **do**
6:         $m_{n,t} \leftarrow \nabla_x \mathcal{L}(f(x + \delta_{n,t-1}), y) + \gamma \cdot m_{n,t-1}$
7:         $\delta_{n,t} \leftarrow \delta_{n,t-1} + \alpha \cdot \text{sign}(m_{n,t})$
8:     **end for**
9: **end for**
10: $m_0 \leftarrow \frac{1}{N} \sum_{n=1}^N m_{n,T'}$
11: **for** t=1 to T **do**                   ▷ The $\{\delta_{k,0}^{dual}\}_{k=1}^K$ are periodically re-initialized
12:     **for** k=1 to K **do**
13:         $g_{k,t} \leftarrow \nabla_x \mathcal{L}(f(x + \delta_{k,t-1}^{dual}), y)$
14:         $\delta_{k,t}^{dual} \leftarrow \delta_{k,t-1}^{dual} + \alpha_t \cdot \text{sign}(g_{k,t})$
15:     **end for**
16:     $m_t \leftarrow \frac{1}{N} \sum_{n=1}^N g_{k,t} + \gamma \cdot m_{t-1}$
17:     $\delta_t \leftarrow \delta_{t-1} + \alpha_t \cdot \text{sign}(m_t)$
18: **end for**
19: **return** $\delta_T$

---

To address this issue, we posit that initialization of the global momentum warrants a thorough examination of the entire surrounding region. We randomly sample several samples in the $\epsilon$-neighborhood of input image $x$ to accumulate the momentum as the global momentum, denoted as randomized global initialization (RGI). By incorporating RGI, we aim to *capture a more representative global momentum that takes into account the diverse local optima surrounding the initial benign sample.*

Lines 1–10 in Alg. 1 outlines the implementation details of random global momentum initialization. Given a benign sample $x$ with its corresponding ground-truth label $y$, we initialize $N$ random perturbations. Each perturbation is added to a separate copy of the benign sample, resulting in $N$ parallel perturbed copies. We then apply the MI-FGSM attack to each perturbed copy for a pre-defined number of iterations $T'$. During this process, we calculate the global momentum achieved in each MI-FGSM run and compute the average global momentum as the enhanced global momentum. Afterward, we reset the perturbation to zero, set the momentum as the enhanced global momentum, and proceed with the adversarial attack using the enhanced global momentum in the subsequent iterations.

**Dual Example**. While RGI is introduced to capture global momentum for initialization, we further propose the dual example strategy to explore a broader loss landscape during the attack process, thereby capturing the global optimization direction more effectively. Unlike previous approaches that explore multiple distinct points around the adversarial example at each step, we amplify the exploration region by sampling more continuous trajectories. In our strategy, each trajectory represents an independent and parallel instance of a dual example, allowing the adversarial example to be optimized across multiple trajectories simultaneously.

In detail shown in line 10–18 of algorithm 1, for an adversarial example $x_{adv}$ to optimize, we first randomly generate $N$ perturbations $\{\delta_n\}_{n=1}^N$ independently, draw from the Gaussian distribution and clip them to the perturbation budget $\epsilon$. Then, we optimize the dual example by I-FGSM in line 12–15, which continuously collect diverse gradients to explore a broader loss landscape. Next, we average the collected gradients and apply them to the update policy of the main adversarial example to optimize.

*Increasing step size.* As shown in fig. 1 and fig. 2, we can notice that all three attacks will converge to the local optima with converging adversarial transferability when increasing the number of iterations. To further study the impact the gradients around the benign sample on the adversarial transferability, we adjust the step size to $\epsilon/T$, where $T = 1, 2, .., 20$, and reproduce the experiments in fig. 1. With small $T$, the utility of gradients near the benign sample and the ability to escape from the local optima far away are improved. As shown in fig. 3, while VMI-FGSM significantly improves adversarial transferability with the help of neighbor information, both I-FGSM and GIMI-FGSM, which rely more the pure gradients, shows a degration with large iterations. It indicates that the importance of near-sample gradients in crafting transferable adversarial examples.

To enhance the capacity of attacks and avoid getting stuck in local optima, the dual example should accumulate gradients that are beneficial for the long-term optimization of adversarial examples. We propose

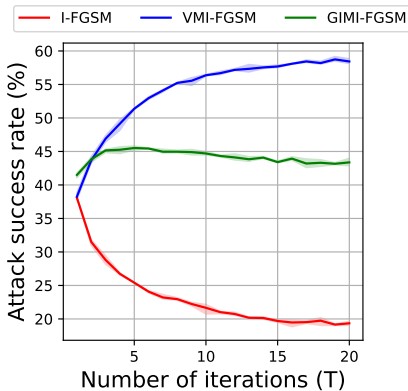

Figure 3: Results of the attack success rate (ASR) versus the epoch for I-FGSM, VMI-FGSM, and GIMI-FGSM with varying step size.

incorporating an increasing step size and a restart mechanism into the dual example strategy. Instead of using a fixed step size, the increasing step size can be more efficiently to sample more gradients near the benign samples , thereby improving transferability. The restart mechanism is designed to generate more trajectories around the benign sample, allowing for the collection of more transferable gradients.

## 4 EVALUATIONS

### 4.1 EXPERIMENT SETUP

**Datasets and models**. In our default setting, we randomly choose $1,000$ images from the ImageNet-1K dataset (Deng et al., 2009) as our evaluation set. We use eight surrogate/victim models, comprising 1) Convolutional Neural Network (CNNs): ResNet-18 (He et al., 2016), ResNet-101 (He et al., 2016), ResNeXt-50 (Xie et al., 2017), DenseNet-121 (Huang et al., 2017), and MobileNet (Howard et al., 2017); and 2) transformers: ViT (Dosovitskiy et al., 2020) and Swin (Liu et al., 2021). We set the surrogate models as ResNet-18, DenseNet-121, and ViT, and evaluate their performance by reporting the mean attack success rate against a series of victim models.

**Baseline methods and implementations**. We apply our proposed randomized global initialization (RGI) and dual example strategy to the VMI-FGSM for adversarial example generation. Since our method is gradient-based, we select a range of state-of-the-art gradient-based attack methods to compare with. These methods are: MI-FGSM (Dong et al., 2018), EMI-FGSM Wang et al. (2021b), VMI-FGSM (Wang and He, 2021), MIG (Ma et al., 2023), PGN (Ge et al., 2023), GIMI-FGSM (Wang et al., 2024), DTA (Yang et al., 2023), and Anda (Fang et al., 2024). To further validate the scalability of our method, we integrate our method as well as other gradient-based methods to the state-of-the-art input transformation-based mehtod SIA (Wang et al., 2023b). To validate the effectiveness of our proposed strategies, we integrate the randomized global initialization, dual example, and decreasing step size (log) to the VMI-FGSM, where the dual example is optimized by MI-FGSM continuously with 5 as the number of epochs to restart.

**Hyper-parameters**. We set the maximum perturbation magnitude $\epsilon = \frac{16}{255}$ under the $L_\infty$ constraint. We set the number of iterations as 10, the step size as $\frac{1.6}{255}$, momentum decay factor $\gamma$ as 1, the look-ahead factor for NI-FGSM as $\frac{1.6}{255}$, the number of additional samples used in EMI-FGSM and VMI-FGSM as 11 and 20, the number of pre-computing epochs for GIMI-FGSM as 5. The balanced coefficient and number of samples for variance estimation in PGN are set as 0.5 and 20, respectively. The order of the approximation of the integral in MIG is set as 20. The relative value for the neighborhood and decay factor for the gradient update in DTA are set as 1.5 and 0.8, respectively. In our method, we set the number of samples for computing the global momentum and dual examples as 5 and 20, respectively. We use the ln sequence as the scheduled increasing step size.

### 4.2 ATTACK A SINGLE MODEL

We first evaluate the effectiveness of our proposed method under the setting of attacking a single model. Specifically, we use different attack methods to generate adversarial examples on three surrogate models: *i.e.*, ResNet-18, DenseNet-121, and ViT. We use one surrogate model at a time. We then evaluate the adversarial transferability of the generated adversarial examples on the eight victim models: ResNet-18, RestNet-101, ResNeXt-50, DenseNet-121, ViT, PiT, Visformer, and Swin. We demonstrate the success rate averaged over the samples separately generated by the three surrogate models intable 1.

As shown in table 1, our proposed method achieves state-of-the-art performance in attacking all models. Specifically, compared to the runner-up method, PGN, which penalizes the gradient norm on the original loss function, our method more efficiently leverages the gradients near the benign samples, resulting in an improvement in adversarial transferability of up to 4.2% against ResNet-101, 5.4% against PiT and Swin, and 3.0% on average. It is worth noting that while PGN focuses on utilizing gradients at each step, our proposed method, including RGI and DE, focuses on the continuous first few steps, where the results demonstrate the superiority of our strategy.

### 4.3 INTEGRATION TO INPUT TRANSFORMATION-BASED METHODS

We then evaluate the compatibility of our method. We integrate the state-of-the-art input transformation-based method structure invariant attack (SIA) into different adversarial

Table 1: Average success rates (%) of attacking eight deep neural networks using various attack methods. The results are averaged over the samples generated using the three separate surrogate models. For simplicity, we denote ResNet-18 as RN18, ResNet-101 as RN101, ResNeXt-50 as RX50, DenseNet-121 as DN121, and Visformer as Vis.

| Method | RN18 | RN101 | RX50 | DN121 | ViT | PiT | Vis | Swin | Avg. |
|--------|------|-------|------|-------|------|------|------|------|------|
| MI | 72.1 | 36.8 | 41.2 | 61.5 | 42.8 | 28.2 | 35.7 | 43.6 | 45.2 |
| EMI | 87.2 | 49.7 | 53.4 | 78.8 | 18.1 | 28.0 | 42.1 | 47.9 | 50.7 |
| VMI | 80.7 | 53.6 | 57.5 | 75.6 | 49.7 | 43.4 | 52.4 | 58.5 | 58.9 |
| GIMI | 80.2 | 44.9 | 49.1 | 71.3 | 44.4 | 33.1 | 43.1 | 50.6 | 52.1 |
| MIG | 79.1 | 48.5 | 53.2 | 74.3 | 47.5 | 38.6 | 46.5 | 55.7 | 55.4 |
| PGN | 89.5 | 65.9 | 69.3 | 86.4 | 53.5 | 53.3 | 62.4 | 69.2 | 68.7 |
| DTA | 77.1 | 44.4 | 49.0 | 68.6 | 45.0 | 31.4 | 42.7 | 50.1 | 51.0 |
| Anda | 83.0 | 62.0 | 66.0 | 83.0 | 53.1 | 50.4 | 61.2 | 63.8 | 65.3 |
| **Ours** | **90.1** | **70.1** | **73.7** | **87.7** | **59.1** | **58.3** | **67.9** | **74.6** | **72.7** |

Table 2: Average success rates (%) of attacking eight deep neural networks using various gradient-based attack methods integrated with SIA. The results are averaged over the samples generated using the three separate surrogate models.

| Method | RN18 | RN101 | RX50 | DN121 | ViT | PiT | Vis | Swin | Avg. |
|--------|------|-------|------|-------|------|------|------|------|------|
| MI | 84.9 | 64.7 | 69.0 | 83.5 | 49.1 | 51.4 | 62.1 | 66.5 | 66.4 |
| EMI | 90.6 | 70.8 | 75.5 | 88.6 | 50.6 | 56.5 | 68.0 | 72.4 | 71.6 |
| VMI | 89.9 | 77.4 | 79.8 | 89.7 | 59.2 | 64.3 | 74.6 | 78.4 | 76.7 |
| GIMI | 92.3 | 74.5 | 79.3 | 90.8 | 52.8 | 59.0 | 71.7 | 74.5 | 74.4 |
| MIG | 90.4 | 75.5 | 78.6 | 90.0 | 58.2 | 62.6 | 72.9 | 76.5 | 75.6 |
| PGN | 94.6 | 79.2 | 83.0 | 92.9 | 57.5 | 64.4 | 74.1 | 78.2 | 78.0 |
| DTA | 93.5 | 82.6 | 85.3 | 93.0 | 57.4 | 66.3 | 79.5 | 80.0 | 79.7 |
| Anda | 90.9 | 79.0 | 82.7 | 90.9 | 60.4 | 64.9 | 76.4 | 78.0 | 77.9 |
| **Ours** | **95.7** | **87.5** | **86.4** | **95.0** | **64.3** | **70.5** | **85.6** | **87.1** | **84.0** |

attack methods. Following the setting in Wang et al. (2023b), we set the number of shuffled copies as 20 and the number of blocks as 3. Other settings during the attack process are aligned with the aforementioned experiments.

The results in table 2 demonstrate that integrating SIA into various adversarial attack methods significantly improves adversarial transferability across all tested models. Our proposed method achieves the highest average success rate of 84.0%, outperforming all other approaches. Compared to existing state-of-the-art methods like DTA and PGN, our method provides substantial improvements, with gains of up to 4.9% in average attack success rates. The largest improvements are seen in transformer-based models, with a 7.1% increase on Swin and a 4.2% increase on PiT, where traditional gradient-based methods tend to struggle. This demonstrates the effectiveness of our strategy in handling both CNNs and vision transformers, making it a powerful tool for adversarial transferability in various model architectures. These results solidify the scalability and superior performance of our method.

### 4.4 EVALUATION UNDER THE ENSEMBLE SETTING

Under the ensemble setting of the pool of three surrogate models, we use different methods to generate the adversarial examples and fool vanilla models as well as advanced defense methods, including adversarial training (AT) (Madry et al., 2018), high-level representation guided denoiser (HGD) (Liao et al., 2018), random smoothing (RS) (Cohen et al., 2019), and neural representation purification (NRP) (Naseer et al., 2020).

We report the results in table 3. In attacking vanilla models under the ensemble setting, our proposed method consistently achieves state-of-the-art performance, outperforming the

Table 3: Average success rates (%) of attacking eight deep neural networks using the adversarial examples crafted by various gradient-based methods using three surrogate models under the ensemble setting.

| Method | RN101 | RX50 | DN121 | PiT | Vis | Swin | NRP | RS | HGD | AT | Avg. |
|--------|-------|------|-------|-----|-----|------|-----|-----|-----|-----|------|
| MI | 59.5 | 63.7 | 85.7 | 46.0 | 57.6 | 63.0 | 36.1 | 23.7 | 53.4 | 33.7 | 52.2 |
| EMI | 80.4 | 83.5 | 95.6 | 66.3 | 78.4 | 81.5 | 50.8 | 27.3 | 73.9 | 37.0 | 67.5 |
| VMI | 75.9 | 78.9 | 92.8 | 63.7 | 72.8 | 75.9 | 52.8 | 27.7 | 70.1 | 36.6 | 64.7 |
| GIMI | 71.4 | 74.7 | 92.8 | 55.4 | 69.2 | 71.2 | 44.7 | 25.9 | 65.6 | 36.0 | 60.7 |
| PGN | 87.8 | 89.2 | 98.6 | 76.9 | 83.6 | 87.9 | 53.5 | 29.4 | 72.7 | 39.9 | 72.0 |
| MIG | 75.2 | 79.9 | 95.4 | 64.2 | 73.8 | 78.4 | 64.6 | 35.8 | 86.0 | 47.5 | 72.1 |
| DTA | 70.2 | 72.8 | 90.4 | 51.2 | 64.6 | 69.3 | 36.2 | 23.0 | 62.4 | 33.8 | 57.4 |
| Anda | 87.6 | 89.4 | 98.6 | 77.7 | 84.3 | 85.2 | 57.5 | 28.0 | 88.0 | 37.7 | 73.4 |
| **Ours** | **90.1** | **90.6** | **99.5** | **82.3** | **86.2** | **88.9** | **65.3** | **38.2** | **89.5** | **48.6** | **77.7** |

runner-up method PGN by a margin of 1.9%. When targeting defense models, our method achieves the highest attack success rate of 38.2% against the most robust defense method, RS. This demonstrates the effectiveness of our approach, not only in attacking standard models but also in overcoming advanced defense mechanisms.

### 4.5 ABLATION STUDY AND DISCUSSION

Table 4: Average attack success rate comparison for different momentum-based attacks. The left subtable presents results for global momentum initialization (GI and RGI), while the right subtable shows the success rate when applying dual example with/without ensemble strategy.

(a) Results of momentum-based attacks integrated with GI or RGI.

|  | MI | NI | PI | EMI | VMI |
|------|------|------|------|------|------|
| Ori. | 62.1 | 63.7 | 65.6 | 69.2 | 76.5 |
| GI | 67.6 | 63.6 | 66.6 | 75.1 | 77.5 |
| RGI | **70.5** | 70.6 | 72.9 | 77.6 | 83.4 |

(b) Attack success rate with none (K=0), single (K=1) and multiple (K=5) dual examples.

| K | I | MI | PI | VMI | GIMI |
|---|------|------|------|------|------|
| 0 | 41.4 | 62.1 | 65.6 | 76.5 | 67.6 |
| 1 | 52.8 | 64.5 | 67.6 | 79.6 | 70.9 |
| 5 | 67.3 | 69.2 | 70.2 | 80.5 | 74.1 |

**On the effect of random global momentum**. Tab. 4a presents the results of random global momentum initialization. It can be observed that global initialization has a minor or negative effect on the adversarial transferability of a few baselines, including NI-FGSM (Lin et al., 2020), PI-FGSM (Gao et al., 2020), and VMI-FGSM. In contrast, the RGI method significantly improves the adversarial transferability for all the baselines, surpassing the GI method with a mean attack success rate of 4.92%. These results provide further confidence in supporting our argument that proper initialization of the global momentum requires a comprehensive exploration of the neighborhood. The effectiveness of the RGI method in enhancing the adversarial transferability across various baselines demonstrates the importance of initializing the momentum in a way that encourages more effective directions.

**On the effect of dual example strategy**. The dual example strategy is plug-and-play, easily integrating into multiple existing attack methods to achieve further performance improvements. To demonstrate its scalability, we integrate it into I- (Goodfellow et al., 2015), MI-, PI-, VMI-, and GIMI-FGSM, using these enhanced attack methods to generate adversarial examples on ResNet-18 and attack other models. The mean attack success rate against the victim models is used as the metric for evaluating adversarial transferability. The results, presented in Tab. 4b, demonstrate clear improvements over the baseline methods. Compared to the baselines, our dual example approach achieves significant performance gains on ResNet-18. Specifically, we observe improvement margins of 25.9% on I-FGSM, 7.5% on MI-FGSM, 4.3% on VMI-FGSM, and 6.9% on GIMI-FGSM. These results further demonstrate the effectiveness of the dual example and highlight the importance of the exploration of the loss landscape in attacks to enhance the adversarial transferability.

**On the use of scheduled step size**. In our proposed method, we incorporate dual examples with an increasing log sequence as the scheduled step to fully exploit the gradients near the benign sample and bypass local optima, thereby enhancing adversarial transferability. To investigate the impact of different sequences on transferability, we conducted experiments, and the results are shown in table 5. Compared to using a constant step size, the scheduled step sequence significantly improves adversarial transferability. Notably, different base attacks benefit from different scheduled steps: log for MI, linear for PI, and VMI, highlighting the importance of choosing the optimal step schedule for each attack method and the necessity to fully utilize the gradient near the benign samples to boost the adversarial transferability.

Table 5: Average attack success rates (%) of classical attack methods when applying dual example with $K = 5$ using different sequences to schedule step size.

| Sequence | I | MI | PI | VMI | GIMI |
|---|---|---|---|---|---|
| constant | 67.3 | 69.2 | 70.2 | 80.5 | 74.1 |
| log | 68.5 | 69.6 | 70.9 | 81.9 | 74.3 |
| linear | 67.7 | 69.1 | 71.5 | 83.0 | 74.5 |
| exp | 69.9 | 69.3 | 70.2 | 79.9 | 73.3 |

## 5 CONCLUSION

In this work, we study the problem of randomness and local optima in adversarial transferability. By leveraging t-SNE to project the optimization trajectory into a low-dimensional space, we observe that while random initialization of adversarial perturbations has little impact on adversarial transferability, the optimization trajectories vary significantly. Motivated by this observation, we propose a randomized global initialization and the use of dual examples to explore more diverse trajectories, enabling the method to overcome multiple optima for improved performance. Extensive experiments on ImageNet-1K demonstrate that our method achieves state-of-the-art results.

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
