# OpenReview forum: "Enhancing Adversarial Transferability Through Exploiting Multiple Randomized Trajectories for Better Global Guidance"
_ICLR.cc/2025/Conference — Submitted to ICLR 2025_

### Official Review · Reviewer_Sa2L · 2024-10-31

**Soundness:** 3
**Presentation:** 3
**Contribution:** 3
**Rating:** 5
**Confidence:** 4

**Summary:**

This paper presents an innovative approach to enhance adversarial transferability by introducing two primary strategies: Randomized Global Initialization (RGI) and Dual Examples. The RGI strategy leverages multiple random perturbations around an initial sample to create a more representative global momentum, thus broadening the optimization landscape and reducing the likelihood of adversarial samples being trapped in local optima. Meanwhile, the Dual Examples strategy generates parallel trajectories for adversarial optimization, effectively exploring a larger portion of the loss landscape and further enhancing transferability. Experimental results on the ImageNet-1K dataset demonstrate that this approach significantly improves attack success rates across various models, including CNNs and vision transformers, underscoring the proposed method's efficacy in increasing adversarial transferability.

**Strengths:**

First, this paper provides a novel approach with meaningful technical contributions and well-supported experimental validations. The introduction of Randomized Global Initialization (RGI) and Dual Examples demonstrates a unique perspective in enhancing adversarial transferability, which could inspire further research in adversarial robustness. Additionally, the paper is well-organized and readable, with detailed descriptions that make complex technical methods accessible. Extensive experimental results on multiple models and datasets, including both CNNs and vision transformers, reinforce the method’s general applicability and strengths in adversarial attack scenarios. The main technical contributions of this paper include the following:
1.	The RGI technique formalizes an approach to initialize adversarial examples across multiple random perturbations, capturing a more representative global momentum for better generalization and reduced local optima entrapment.
2.	The Dual Example Strategy enhances the transferability of adversarial examples by generating parallel trajectories, effectively exploring a larger portion of the loss landscape. This broad approach ensures more robust adversarial optimization across different models.
3.	The proposed RGI and Dual Examples are seamlessly integrated with existing gradient-based methods, highlighting the flexibility and adaptability of the proposed approach across various adversarial attack frameworks.
4.	Extensive experiments on the ImageNet-1K dataset demonstrate that the method outperforms other adversarial transfer techniques. The paper provides theoretical insights that underscore the importance of initialization and trajectory exploration in adversarial attacks, contributing to the broader understanding of optimization in high-dimensional spaces.

**Weaknesses:**

1.	The Randomized Global Initialization and Dual Example strategies introduce significant computational requirements, especially when optimizing multiple trajectories simultaneously. This could limit the method's practicality in resource-constrained environments.
2.	The approach relies on several hyperparameters (e.g., number of random initializations, step size sequence), which may require fine-tuning for different models. This sensitivity could hinder straightforward application and scalability across diverse model architectures.
3.	This article may lack a theoretical proof for the validity of the global initialization. In addition, there is a lack of experimental proof of the optimal settings for the number of samples to compute the global momentum and dual examples.

**Questions:**

1.	The introduction of randomized global initialization and dual samples strategies increases the computational overhead. Does the paper quantitatively evaluate the time complexity and computational resource requirements of these strategies? How computationally efficient is this approach in practical applications?
2.	The method is sensitive to some hyperparameters in different models; have the authors evaluated the optimal values for these parameters on different models? How are these parameters chosen in practical applications?
3.	The paper primarily compares its method with gradient-based approaches but does not address non-gradient-based adversarial attack methods. What are the advantages and disadvantages of this method compared to non-gradient-based approaches?
4.	Is there an issue with adversarial example stability due to certain random initializations when using randomized global initialization? Has the author evaluated the variance in attack success rates across different random initializations?

---

### Official Review · Reviewer_eAAG · 2024-11-01

**Soundness:** 2
**Presentation:** 3
**Contribution:** 2
**Rating:** 5
**Confidence:** 4

**Summary:**

The paper introduces two key strategies—randomized global initialization (RGI) and dual example generation (DE)—which leverage multiple optimization trajectories to explore the loss landscape more comprehensively. This is a novel addition to adversarial attack literature, aiming to improve transferability by avoiding local optima.

**Strengths:**

1. It is necessary to dive into transferable attacks to discover hidden defects of neural networks, especially for realistic black-box environments.

2. The authors conducted extensive experiments, reporting performance metrics across a diverse set of models (ResNet-18, DenseNet-121, ViT, etc.) and testing both single-model and ensemble settings. The results consistently show improvement in attack success rates, particularly for transformer-based models.

**Weaknesses:**

**Main concerns:**

**1. Concerns about innovation:**

The generation method of Dual Examples is still unclear. In line 293, the author claims to use random Gaussian noise to generate N initialization samples. Are the sampling methods for random initialization of momentum and dual samples consistent? If so, there may be some overlap in the sampling methods. The random perturbation sampling in momentum initialization (such as randomly initializing multiple perturbations) is basically consistent with the generation of Dual Examples (sampling multiple trajectories from the neighborhood), both of which are sampled in the neighborhood and optimized on their own independent trajectories. This means that Dual Examples actually overlaps with the strategy of momentum initialization, does not really provide new information or optimization paths, and only increases the complexity of the calculation. Could I ask the authors to provide a detailed comparison of the sampling methods used for random initialization of momentum and dual samples?

In addition, since there are already numerous works **[1] [2] [3]** that combine the gradient information of neighborhood samples to improve transferability, could I think that the core of this paper is essentially combine neighborhood exploration (through random initialization and Dual Examples) with the pre-convergence momentum strategy of GIMI-FGSM? The pre-convergence momentum strategy has been reflected in GIMI-FGSM, and more gradient information is introduced by neighborhood increasing exploration (random initialization and Dual Examples) to calculate the average momentum, mainly by sampling multiple examples in the neighborhood. Could I ask the authors to provide a more detailed comparison of their method with existing works, particularly focusing on how their approach differs from or improves upon the combination of neighborhood exploration and pre-convergence momentum strategies?

**2. Randomized Initialization Without Sufficient Parameter Analysis:**

The paper proposes randomized global initialization but does not provide a systematic study on how different levels of randomness affect convergence and transferability. Specifically, there is no ablation to explore the sensitivity of RGI to the number of random initializations or perturbation magnitude.

RGI uses a predefined number of samples, yet the impact of this parameter **N**  remains unclear. Testing different sample sizes or introducing an analysis of the trade-offs between computation cost and performance gain would make the method more practical and understandable.

**3. Vagueness on Empirical Validation:**

While the experimental results are promising, the paper’s reliance on empirical data without deeper technical analysis limits the work’s robustness. For instance, t-SNE visualizations show trajectories across random initializations but fail to address how these trajectories relate to transferable gradient directions in high-dimensional space.

The contribution of Figure 2 is ambiguous. In lines 214-215, the author says "running GIMI-FGSM from different random starting points often causes adversarial examples to converge to distinct local optima", but Figure 2 is only a visualization of adversarial sample updates and does not reflect the concept of "local optimum". In addition, the author claims in lines 197-198 that "even with the same step size and number of optimization steps, each attack pushes the adversarial example to different distances from the benign sample." Obviously, when the input perturbations are inconsistent, the update directions of the adversarial samples generated by random initialization are different. This phenomenon does not explain the contribution of random initialization to transferability. I suggest modifying Figure 2 to more clearly reflect the motivation.

**4. Computational overhead of multiple trajectories:**

The core method of this paper relies on multi-trajectory optimization of adversarial examples, including random initialization and Dual Examples, which means that each update requires separately calculating gradients on multiple trajectories. This process significantly increases the computational cost because each trajectory needs to be forward and backward propagated independently, and then the gradient information of different trajectories is integrated for update. This multiple optimization trajectories increase the demand for computing resources and memory to a certain extent. Especially on large-scale models or datasets (such as ImageNet), such consumption may not be negligible. Comparing the inference time of the proposed method with other baselines can effectively evaluate the efficiency of the algorithm.

**References:**

**[1]** Wang, X., & He, K. (2021). Enhancing the transferability of adversarial attacks through variance tuning. In Proceedings of the IEEE/CVF conference on computer vision and pattern recognition (pp. 1924-1933).

**[2]** Zhu, H., Ren, Y., Sui, X., Yang, L., & Jiang, W. (2023). Boosting adversarial transferability via gradient relevance attack. In Proceedings of the IEEE/CVF International Conference on Computer Vision (pp. 4741-4750).

**[3]** Wang, X., Jin, Z., Zhu, Z., Zhang, J., & Chen, H. (2024, October). Improving Adversarial Transferability via Frequency-Guided Sample Relevance Attack. In Proceedings of the 33rd ACM International Conference on Information and Knowledge Management (pp. 2410-2419).


**Minor concerns:**

**1. Ambiguity of pseudocode parameters：**

The value of the **T'** parameter in line 5 of the pseudocode is not specified. Since its function is similar to that of the parameter **P** in GIMI-FGSM, can it be assumed that it is set to 5 with reference to the parameter selection of GIMI-FGSM? Could I ask the authors to clarify the value of **T'** and explain its relationship to the **P** parameter in GIMI-FGSM?

**2. Possible typo errors in pseudocode:**

In line 16 of the pseudocode, should **$\frac{1}{N} \sum_{n=1}^{N}$** be **$\frac{1}{K} \sum_{k=1}^{K}$**? I'm not sure. **$g_{k,t}$** is based on the gradient of **K** Dual Examples, so **1/K** should be used instead of **1/N** when averaging **$g_{k,t}$** (here **N** is the number of samples used to calculate the randomly initialized momentum, and it has ended the loop in line 9).

**3.Reproducibility Concerns:**

Given the complexity of the proposed strategies and the lack of specific initialization parameters, reproducibility may be challenging for future researchers. If possible, open-sourcing the code would help improve transparency, allowing the community to validate and build upon the results.

**Questions:**

Please refer to the **Weakness section**. I might raise my score if the authors address my concerns, especially regarding the computational overhead of the algorithm.

---

### Official Review · Reviewer_qyKe · 2024-11-04

**Soundness:** 1
**Presentation:** 2
**Contribution:** 3
**Rating:** 3
**Confidence:** 4

**Summary:**

This paper introduced new optimization strategies for adversarial attacks called Randomized Global Initialization and Dual Example. These methods trade-off computational cost for improved transferability by exploring more of the loss landscape. The authors demonstrated through extensive experiments that Randomized Global Initialization and Dual Example significantly boost the performance of gradient-based attack methods by leveraging broader optimization paths.

**Strengths:**

**1.** This paper has well orgnized visualization, which effectively helps theoretical derivation. For example, Figure 2 clearly explains the different paths of the FGSM during the iterative process.

**2.** The novel method achieves SOTA results in the major experiments, which aligns with those inferences.

**Weaknesses:**

**1.** The authors claim that their methods enhances the adversarial transferability of attacks, but does not conduct enough evaluation under defences to prove this.
For example, some novel adversarial defence methods claim that they can defend the attackers by reducing adversarial transferability [1,2,3,4]. If these strong adversarial defence algorithms could be used as a benchmark and given the success rate of the attack, it would better demonstrate the validity of the advantage in adversarial transferability.

**2.** The authors don't seem to mention the limitations of their paper.

[1] G. Carbone et al., “Robustness and interpretability of neural networks’ predictions under adversarial attacks,” 2023.

[2] Y. Ma, M. Dong, and C. Xu, “Adversarial robustness through random weight sampling,” in Advances in Neural Information Processing Systems, A. Oh, T. Naumann, A. Globerson, K. Saenko, M. Hardt, and S. Levine, Eds., vol. 36. Curran Associates, Inc., 2023, pp. 37 657–37 669.

[3] M. Dong, X. Chen, Y. Wang, and C. Xu, “Random normalization aggregation for adversarial defense,” Advances in Neural Information Processing Systems, vol. 35, pp. 33 676–33 688, 2022.

[4] B. Li, C. Chen, W. Wang, and L. Carin, “Certified adversarial robustness with additive noise,” Advances in neural information processing systems, vol. 32, 2019.

**Questions:**

see weaknesses

---

### Official Review · Reviewer_Vn5x · 2024-11-04

**Soundness:** 2
**Presentation:** 1
**Contribution:** 2
**Rating:** 3
**Confidence:** 3

**Summary:**

This paper addresses the challenge of enhancing adversarial transferability in deep neural networks (DNNs) by proposing new strategies to avoid local optima during the generation of adversarial examples.

**Strengths:**

(1) The paper is well-structured.

(2) The research topic is significant.

**Weaknesses:**

(1) The novelty is limited. I believe that the essence of the proposed AGI and dual examples (DE) are equivalent to reducing the variance of attack directions, since RGI and DE accumulated/averaged multiple attack directions for each perturbation updating.  While accumulating multiple attack directions for stabling each perturbation updating has been proposed in [1][2].

(2) Insufficient Evaluation: The evaluations presented are not robust enough. Given the similarity of the proposed approach to methods in [1][2], it is crucial to include these as baseline comparisons. Moreover, widely recognized transferable attacks such as DIM [3] and TIM [4] should also be included as baselines

(3) The attack success rates reported in Table 3 against defense methods like NRP, RS, HGD, and AT are notably low. In contrast, prior methods like DIM and TIM have achieved higher success rates against these defenses, raising concerns about the fairness and validity of the evaluation.

(4) Since AGI and DE introduce additional steps in generating perturbations, it is unfair to compare the proposed methods and baselines with differing numbers of optimization steps.


(5) typos and format errors:
    (1) In abstraction, line 4,  "samplesoften"  (2) in Section 2.2, the reference format is not correct.


[1] Wu, Lei, Zhanxing Zhu, and Cheng Tai. "Understanding and enhancing the transferability of adversarial examples." arXiv preprint arXiv:1802.09707 (2018).

[2] Huang, Tianjin, et al. "Direction-aggregated attack for transferable adversarial examples." ACM Journal on Emerging Technologies in Computing Systems (JETC) 18.3 (2022): 1-22.

[3] Xie, Cihang, et al. "Improving transferability of adversarial examples with input diversity." Proceedings of the IEEE/CVF conference on computer vision and pattern recognition. 2019.

[4] Dong, Yinpeng, et al. "Evading defenses to transferable adversarial examples by translation-invariant attacks." Proceedings of the IEEE/CVF conference on computer vision and pattern recognition. 2019.

**Questions:**

See weakness.

---

### Meta-Review · Area_Chair_wxx7 · 2024-12-21

**Metareview:**

This paper studies transferable adversarial examples. Its key contributions include two technical strategies: randomized global initialization and dual examples, which create multiple attack trajectories for enhancing the optimization of adversarial examples. Empirical results on ImageNet are provided to support the effectiveness of the developed attack.

While the reviewers found this paper interesting to read, they raise multiple significant concerns about this paper, including limited novelty and insufficient ablations. However, no rebuttal is provided for addressing these concerns. The final decision is rejection.

**Additional Comments On Reviewer Discussion:**

No rebuttal is provided and all reviewers have ratings below 5.

---

### Decision · Program_Chairs · 2025-01-22

Reject